# Cationic Chitooligosaccharide Derivatives Bearing Pyridinium and Trialkyl Ammonium: Preparation, Characterization and Antimicrobial Activities

**DOI:** 10.3390/polym15010014

**Published:** 2022-12-20

**Authors:** Conghao Lin, Zhanyong Guo, Aili Jiang, Xiaorui Liang, Wenqiang Tan

**Affiliations:** 1Yantai Institute of Coastal Zone Research, Chinese Academy of Sciences, Yantai 264003, China; 2College of Life Sciences, Yantai University, Yantai 264005, China; 3School of Basic Sciences for Aviation Naval Aviation University, Yantai 264001, China

**Keywords:** chitooligosaccharide, nicotinylation, pyridinium cation, antibacterial, antifungal

## Abstract

In this study, chitooligosaccharide-niacin acid conjugate was designed and synthesized through the reaction of chitooligosaccharide and nicotinic acid with the aid of *N*,*N*′-carbonyldiimidazole. Its cationic derivatives were prepared by the further nucleophilic substitution reaction between the chitooligosaccharide-niacin acid conjugate and bromopropyl trialkyl ammonium bromide with different alkyl chain lengths. The specific structural characterization of all derivatives was identified using Fourier Transform Infrared Spectroscopy (FTIR) and Nuclear Magnetic Resonance (NMR), and the degree of substitution was obtained using the integral area ratio of the hydrogen signals. Specifically, the antibacterial activities against *Escherichia coli*, *Staphylococcus aureus*, *Pseudoalteromonas citrea* and *Vibrio harveyi* were evaluated using broth dilution methods. In addition, their antifungal activities, including *Botrytis cinerea*, *Glomerella cingulate* and *Fusarium oxysporum f.* sp. *cubense* were assayed in vitro using the mycelium growth rate method. Experimental data proved that the samples showed antibacterial activity against four pathogenic bacteria (MIC = 1–0.125 mg/mL, MBC = 8–0.5 mg/mL) and enhanced antifungal activity (50.30–68.48% at 1.0 mg/mL) against *Botrytis cinerea*. In particular, of all chitooligosaccharide derivatives, the chitooligosaccharide derivative containing pyridinium and tri-n-butylamine showed the strongest antibacterial capacity against all of the test pathogenic bacteria; the MIC against *Vibrio harveyi* was 0.125 mg/mL and the MBC was 1 mg/mL. The experimental results above showed that the introduction of pyridinium salt and quaternary ammonium salt bearing trialkyl enhanced the antimicrobial activity. In addition, the cytotoxicity against L929 cells of the chitooligosaccharide derivatives was evaluated, and the compounds exhibited slight cytotoxicity. Specifically, the cell viability was greater than 91.80% at all test concentrations. The results suggested that the cationic chitooligosaccharide derivatives bearing pyridinium and trialkyl ammonium possessed better antimicrobial activity than pure chitooligosaccharide, indicating their potential as antimicrobial agents in food, medicine, cosmetics and other fields.

## 1. Introduction

Bacteria are ubiquitous in life. Pathogenic bacteria have found in a wide range of products in recent years, which increase the risks of infections causing severe illness in human beings. *Staphylococcus aureus* and *Escherichia coli* are common pathogens that can cause inflammation in the body. Furthermore, with the vigorous development of marine aquaculture, especially the promotion of high-density aquaculture, the quality and yield of seafood are affected by some marine bacterial infections such as *Pseudoalteromonas citri* and *Vibrio harveyi* [1]. In addition, plant pathogenic fungi have a terrible impact on melons, vegetables and other cash crops, causing a lot of economic losses to agriculture. For instance, fruits and vegetables can rot after harvest because of *Botrytis cinerea* infections. *Fusarium oxysporum f.* sp. *cubense* can attack bananas, resulting in wilt and rot. *Glomerella cingulate* can be greatly harmful to fresh fruits and further lead to their harm. More seriously, seed-borne fungal pathogens present externally or internally may cause seedling decay, necrosis and damage to plant growth [2]. Antibiotics and chemical inhibition agents are commonly used to control pathogenic microorganisms. However, the unreasonable use of them may cause resistance to drugs. Moreover, the residues of antibiotics and pesticides also have a negative impact on the environment and humans [3]. Therefore, it is urgent to develop safe, effective and environmentally friendly natural antimicrobial agents. In this context, chitosan and chitooligosaccharides are considered to have multiple functional properties and serve as potential candidates in many industries on account of their excellent biological activities, easy modification and broad application prospects in the medicine, food, agricultural and environmental protection fields.

Chitosan, the deacetylation product of chitin, is a linear polysaccharide. Unfortunately, the shortcomings of chitosan (involving poor solubility and biological activity) affect its further development and application [4]. Chitooligosaccharide (COS) is the degradation product obtained by reducing the polymerization degree of chitosan through chemical, physical and enzymatic degradation. Compared with chitosan, COS exhibits unique physiological activities and functions (antioxidation, antimicrobial, hygroscopic and moisturizing activity) [5,6]. Owning to its distinctive physical and chemical features, COS has been extensively used in agriculture, wastewater treatment, food preservation and other fields [7]. The scope of unmodified COS is incomparable to artificial synthesis compounds owing to their relatively weak biological activity. Luckily, the high amount of free amino and hydroxyl groups in COS supply reaction sites for further reactions [8]. A reasonable strategy to enhance the antimicrobial activity of COS is chemical modification by introducing efficient antimicrobial functional groups onto the COS structure. The modified derivatives not only retain the original benefits and the natural backbone of COS, but also endow them with enhanced antimicrobial properties [9]. Cationic chitosan and COS derivatives have been reported to possess specific antimicrobial, moisture absorption, anti-inflammatory activities, etc. in vitro. Furthermore, they have been widely used as antibacterial agents in cosmetics, food, textiles and other applications [10].

Chitosan pyridinium salt is one of the typical chitosan cationic derivatives. For instance, Rui et al. prepared *N*-(1-carboxybutyl-4-pyridinium) chitosan chloride (pyridine chitosan) and investigated their antifungal behavior. These authors found that the chitosan derivative was a much better fungicidal agent than pure chitosan against the *B. cinerea* and *Fulvia fulva* tested, and they pointed out that the compounds could effectively inhibit the growth of mycelium and the germination of spores [11]. However, there are few reports on the preparation and antimicrobial activity of COS pyridinium salt, and more research needs to explore the effect on the antimicrobial activity of COS derivatives. In addition, the impact of quaternary ammonium salt bearing trialkyl on antimicrobial activities has ignited the interests of investigators. For example, Zhou et al. synthesized and characterized four kinds of chitosan derivatives by introducing quaternary ammonium salt bearing linear alkyl, and they found that the antibacterial activity of the compounds increased with increasing quaternary ammonium salt substitution and carbon chain-length [12]. Based on the results of the experimental analyses above, it is expected that complexing pyridinium salt and quaternary ammonium salt bearing trialkyl onto the structure of COS may result in the development of novel COS derivatives with enhanced antimicrobial activity.

This study aimed to synthesize COS derivatives with efficient antimicrobial activities, which could serve as ideal candidates for use in the development of antimicrobial material in many industries. A series of cationic COS derivatives bearing pyridinium and trialkyl ammonium were prepared via a two-step reaction with nicotinylation and nucleophilic substitution. The specific structural characterization of the COS derivatives was analyzed using Fourier transform infrared spectra (FTIR) and nuclear magnetic resonance (NMR). Moreover, the positive roles of pyridinium salt and quaternary ammonium salt bearing trialkyl on antimicrobial activities were investigated. Antibacterial activities against *Escherichia coli* (*E. coli*), *Staphylococcus aureus* (*S. aureus*), *Pseudoalteromonas citrea* (*P. citrea*) and *Vibrio harveyi* (*V. harveyi*) using broth dilution methods, and antifungal activities against *Botrytis cinerea* (*B. cinerea*), *Glomerella cingulate* (*G. cingulata*) and *Fusarium oxysporum f.* sp. *cubense* (*F. oxysporum*) via mycelium growth rate method were tested and reported. Furthermore, the cytotoxicity of COS and its derivatives against L929 cells was assessed using MTT assay in vitro.

## 2. Materials and Methods

### 2.1. Materials

Chitooligosaccharide lactate (deacetylation degree of 94%, molecular weight ≤ 3000 Da) was obtained from Weikang Biomedical Technology Co., Ltd. (Linyi, China). Dimethyl sulfoxide (DMSO), acetonitrile, acetone, absolute ethanol, ethyl acetate, trimethylamine (33% ethanol solution) and triethylamine were purchased from Sinopharm Chemical Reagent Co., Ltd. (Shanghai, China). Nicotinic acid, *N*,*N*′-carbonyl diimidazole (CDI) and 1,3-dibromopropane were obtained from Macklin Biochemical Co., Ltd. (Shanghai, China). Tripropylamine and tributylamine were purchased from Aladdin Reagent Co., Ltd. (Shanghai, China). All drugs and reagents were obtained from commercial sources and used without further processing.

### 2.2. Characterization Methods

The FTIR spectra of COS and COS derivatives were characterized by a Nicolet iS50 instrument (Thermo Fisher Scientific, Waltham, MA, USA) at the range from 4000–500 cm^−1^ at 25 °C. In each experiment, the sample was ground with KBr in the weight ratio of 1:50 for the measure. Moreover, the ^1^H NMR and ^13^C NMR spectra were determined using a Bruker AVIII-500 Spectrometer (500 MHz, purchased from Bruker Technologies and Services Co., Ltd., Beijing, China) under a static magnetic field at 500 MHz. Chemical shifts were reported in parts per million (ppm) on the *δ* scale. For the ^1^H NMR spectra, 20 mg of COS and its derivatives were dissolved in 0.6 mL of D_2_O and 40 mg of compounds were dissolved in 0.6 mL of DMSO-*d*_6_ for ^13^C NMR spectrogram.

The DS of COS-N and cationic COS derivatives bearing pyridinium and trialkyl ammonium derivatives was quantitatively calculated by the integral area ratio of the hydrogen signals [13]. The DS was calculated by the following equation:(1)$$\mathrm{DS}1\;(\%)=\frac{5I_{H\;\mathrm{pyridine}}}{4I_{H\;3-6}}$$
(2)$$\mathrm{DS}2\;(\%)=\frac{4I_{Hb}}{9I_{Ha}}\times \mathrm{DS}1$$
where I_H 3–6_ is the integral value of the H3–H6 (3.17–4.30 ppm) on the COS backbone, I_H pyridine_ was the integral value of the hydrogen in the pyridine ring (7.16–9.02 ppm) of COS-N, I_Ha_ was the integral value of the hydrogen in the pyridinium salt and I_Hb_ was the integral value of the hydrogen at the end of the alkyl chain.

### 2.3. Chemical Synthesis

#### 2.3.1. Synthesis of Bromopropyl Trialkyl Ammonium Bromides

Bromopropyl trialkyl ammonium bromides (BPTABs) with different alkyl chain lengths were synthesized according to Figure 1. A total of 5.08 mL, 50 mmol of 1,3-Dibromopropane was dispersed in 20 mL of acetonitrile. Then, 50 mmol of trimethylamine ethanol solution (13.57 mL), triethylamine (6.95 mL), tripropylamine (9.48 mL) and tributylamine (11.88 mL) were added in ice bath, respectively. The reaction mixture was stirred at room temperature for 5 h and precipitated with excessive ethyl acetate. Several precipitates were washed with ethyl acetate three times. A total of 4 kinds of BPTABs with different alkyl chain lengths were obtained after freeze-drying in vacuo for 48 h. Compound **1**: FTIR (KBr) v/cm^−1^, 3012, 2966, 1480, 563. ^1^H NMR (500 MHz, DMSO-*d*_6_), *δ* 3.51 (t, 2H), 3.45 (m, 2H), 3.09 (s, 9H), 2.23 (m, 2H), ^13^C NMR (500 MHz, DMSO-*d*_6_), *δ* 62.2, 52.9, 31.16, 25.97. Compound **2**: FTIR (KBr) v/cm^−1^, 2985, 2902, 1486, 558. ^1^H NMR (500 MHz, DMSO-*d*_6_), *δ* 3.65 (m, 2H), 3.30 (m, 6H), 3.22 (t, 2H), 2.18 (m, 2H), 1.20 (t, 9H). ^13^C NMR (500 MHz, DMSO-*d*_6_), *δ* 55.1, 52.6, 31.3, 24.9, 7.7. Compound **3**: FTIR (KBr) v/cm^−1^, 2964, 2878, 1476, 601. ^1^H NMR (500 MHz, DMSO-d6), *δ* 3.62 (m, 2H), 3.34 (t, 2H), 3.21 (t, 6H), 2.20 (m, 2H), 1.63 (m, 6H), 0.88 (t, 9H). ^13^C NMR (500 MHz, DMSO-*d*_6_), *δ* 59.8, 56.9, 31.3, 25.1, 15.3, 10.9. Compound **4**: FTIR (KBr) v/cm^−1^, 2961, 2874, 1467, 558. ^1^H NMR (500 MHz, DMSO-*d*_6_), *δ* 3.46 (t, 6H), 3.31 (m, 2H), 3.20 (t, 2H), 2.20 (m, 2H), 1.62 (m, 6H), 1.32 (m, 6H), 0.94 (t, 9H). ^13^C NMR (500 MHz, DMSO-*d*_6_), *δ* 58.2, 52.1, 31.3, 25.4, 23.5, 19.8, 14.0.

#### 2.3.2. Preparation of Chitooligosaccharide-Niacin Acid Conjugate (COS-N)

The COS-N was performed as described previously [14] with a minor modification. Nicotinic acid (19.8 g, 160 mmol) was dissolved in DMSO (40 mL) with stirring at room temperature. Subsequently, CDI with equal mole (160 mmol) was added slowly, after the reaction with stirring at 60 °C for 12 h under N_2_ atmosphere, followed by mixing with 30 mL of DMSO containing 12.88 g of COS. The reaction mixture was also reacted for 12 h at 60 °C under N_2_ atmosphere. The COS-N was precipitated by adding 400 mL of acetone and then filtered to recover the precipitate. The isolated product was depurated by dissolving with DMSO and precipitating with acetone three times, then washed with anhydrous ethanol several times. Finally, COS-N was obtained by drying overnight in a vacuum freeze dryer.

#### 2.3.3. Preparation of Cationic Chitooligosaccharide Derivatives Bearing Pyridinium and Trialkyl Ammonium

Four kinds of end products were synthesized as described previously [15]. A total of 1.06 g (3 mmol) of COS-N was dissolved completely in 6 mL of DMSO at room temperature. Then, 2.09 g (8 mmol) of bromopropyl trimethyl ammonium bromide, 2.43 g (8 mmol) of bromopropyl triethyl ammonium bromide, 2.76 g (8 mmol) of bromopropyl tripropyl ammonium bromide and 3.10 g (8 mmol) of bromopropyl tributyl ammonium bromide were added into the above COS-N solution, respectively. The mixture was stirred for 24 h at 60 °C under N_2_ atmosphere. The reaction solution was poured into 100 mL of acetone to obtain some crude products. The precipitates were dissolved with DMSO and precipitated with acetone three times, followed by washing with anhydrous ethanol. Finally, four end products were achieved after vacuum freeze-drying for 24 h.

### 2.4. Antibacterial Assays

MIC and MBC were determined for COS and COS derivatives against *E. coli*, *S. aureus*, *P. citrea* and *V. harveyi* by broth dilution methods [16,17]. The COS, COS-N and four end products were dissolved in deionized water at 64 mg/mL. Firstly, the above bacteria were incubated in bacterial liquid medium at 37 °C for 18 h with 120 rpm. A total of 100 µL of deionized water was added to each well of the 96-well cell culture plate, samples solutions were serially diluted to the appropriate concentration, followed by adding 100 μL of bacterial liquid into each well. The tested concentrations were controlled at 16, 8, 4, 2, 1, 0.5, 0.25, 0.125, 0.0625, 0.03125, 0.015625 and 0.078125 mg/mL, respectively. After 18 h of nurturing at 37 °C, the MIC of the samples was determined using the lowest concentration of the medium that was clear and had no bacterial growth. The MBC of the COS and COS derivatives were further determined by incubating the bacterial suspension.

### 2.5. Antifungal Assays

The antifungal activities of the COS and COS derivatives against *B. cinerea*, *G. cingulate* and *F. oxysporum* were tested via the mycelium growth rate method [18]. In short, compounds were dissolved in deionized water at 10 mg/mL. Then, in the sterile environment, 1.5 mL, 0.75 mL, and 0.15 mL of sample solutions were poured into 13.5 mL, 14.25 mL and 14.85 mL of fungi mediums, respectively. The final concentrations of the samples were controlled at 1.0, 0.5 and 0.1 mg/mL. Thereafter, fungi mediums containing samples with different final concentrations were poured into the culture dishes (9.0 cm) for solidification. In addition, a volume of deionized water equal to the sample solution was used as the blank group. Finally, a fungal cake disk of 5.0 mm diameter was inoculated in the center of the medium and cultivated at 27 °C. The diameter of mycelium was measured in millimeters until the fungal hyphae of the control group spread to the edges of the medium. The inhibition index for COS and COS derivatives was calculated according to the following equation:Inhibitory index (%) = [1 − (Da − 5)/(Db − 5)] × 100(3)
where Da was the diameter of the growth zone in the test plates and Db was the diameter of the growth zone in the blank plates.

### 2.6. Cytotoxicity Assay

The cytotoxicity of COS, COS-N and cationic COS derivatives bearing pyridinium and trialkyl ammonium on L929 cells was determined by MTT assay in vitro at different concentrations of 10, 50, 100, 500 and 1000 μg/mL [19]. In brief, the L929 cells were incubated in DMEM-F12 (involving 10% fetal calf serum and 1% mixture of penicillin and streptomycin) and inoculated into 96-well cell culture plates at 37 °C in a 5% CO_2_ cell incubator. Then, the cells were treated for 24 h with culture medium containing compounds with various concentrations. Thereafter, 10 μL of 5 mg/mL MTT solution was added to each well, followed by culturing for 3 h. Culture medium in 96-well cell culture plates was poured out and 100 μL of DMSO was re-added to each well. After shaking for 15 min, the absorbance at 490 nm was measured with the microplate reader. The cell viability was calculated according to the following equation:Cell viability (%) = A compound/A control × 100 (4)
where A compound is the absorbance of the test groups and A control is the absorbance of the control groups.

### 2.7. Statistical Analysis

All measurements were carried out in three replications and expressed as mean ± standard deviation.

## 3. Results and Discussion

In this study, chitooligosaccharide-niacin acid conjugate (COS-N) was synthesized by a nicotinylation reaction under the condition of *N*,*N*′-carbonyl diimidazole (CDI) as the catalyst. Specially, CDI can react with functional carboxylic acid to obtain carbonyimidazole as intermediate, which possesses strong reactivity and can further react with the 2-amino and 6-hydroxyl groups on COS to form amide and ester bonds [20,21]. The cationic derivatives of COS-N were acquired by further reaction between COS-N and BPTABs. One interesting finding was that the pyridinium salt not only acted as the active group to enhance the antimicrobial activity directly but also acted as a bridge to connect the quaternary ammonium salt bearing trialkyl. The chemical structures of products were explained by the FTIR and NMR.

### 3.1. FT-IR Spectra

The FTIR spectra of COS, COS-N and four end products are shown in Figure 1. For pure COS, the characteristic peak that appeared at 3410 cm^−1^ was attributed to the stretching vibrations of NH and OH, the typical peak appeared at 2870 cm^−1^ was assigned to the stretching vibration of methylene, the new peak at 1600 cm^−1^was due to the vibration modes of N-H and the characteristic peak located at 1070 cm^−1^ was assigned to the stretching vibration of the glucosamine ring (C-O) [22]. After nicotinylation, in the COS-N spectrum, the new peaks at 1730 cm^−1^, 1650 cm^−1^ and 1284 cm^−1^ were attributed to the stretching vibration of the ester bond (C=O), the stretching vibration of the amide bond (C=O) and the stretching vibrations of the ester bond (C–O), respectively [23]. A set of peaks at 628–740 cm^−1^ were attributed to the vibrational absorption peak of pyridine (C=C) [24]. These data initially confirmed the successful synthesis of COS-N. After the conjugation of COS-N and BPTABs, the FTIR spectra of 4 end products showed new peaks at 1475 and 1502 cm^−1^, which were assigned to the characteristic absorbance of pyridinium cation and quaternary ammonium salt [25,26]. In summary, the FTIR data preliminarily verified the successful synthesis of cationic COS derivatives bearing pyridinium and trialkyl ammonium. More valid evidence also needs to be provided by NMR.

### 3.2. H NMR Spectra

The chemical structures of unmodified COS and COS derivatives were further elucidated by the ^1^H NMR spectra, and the attributions of characteristic peaks are shown in Figure 2. The proton signals of COS backbone were located in 5.30 ppm (H1), 3.20–4.30 ppm (H3–H6) and 2.98 ppm (H2) [27,28]. After nicotinylation, as shown in the spectrum of COS-N, the characteristic signals at 7.29–9.00 ppm were assigned to the hydrogen atoms on pyridine (a, b, c, d) [29]. For 4 end products, characteristic peaks located at 4.37 ppm, 2.44 ppm, and 3.35 ppm were assigned to the hydrogen atoms on methylene (e, f, g) [30]. In sample A, apart from the chemical shift of the hydrogen atoms on methylene, a new peak at 3.07 ppm was assigned to the hydrogen atoms on the chemical shift of -N^+^(CH_3_)_3_, indicating the successful reaction of bromopropyl trimethyl ammonium bromide with COS-N. In the ^1^H NMR spectrum of compound B, the peaks at 3.22 and 1.17 ppm were associated with the hydrogen atoms (h and i) on -N^+^(CH_2_CH_3_)_3_. In sample C, a set of peaks at 3.10, 1.59 and 0.82 ppm were attributed to the hydrogen atoms on tripropyl (h, i, and j). New peaks at 3.34, 1.54, 1.23 and 0.81 ppm were ascribed to the protons h, i, j and k of -N^+^(CH_2_CH_2_CH_2_CH_3_)_3_ on compound D, respectively [31,32]. In conclusion, the above ^1^H NMR spectrum analysis indicated the successful preparation of cationic chitooligosaccharide derivatives bearing pyridinium and trialkyl ammonium through the nucleophilic substitution reaction of COS-N and BPTABs with different alkyl chain lengths.

### 3.3. C NMR Spectra

The ^13^C NMR spectra was used to further explain the chemical structures of the COS and COS derivatives; the peaks are illustrated in Figure 3. For the pristine COS, the resonances at 101.0, 77.4, 70.6, 66.6, 61.2, 56.8 and 177.6 ppm (C1, C4, C5, C3, C6, C2 and the carbon atom from the undeacetylated part) belonged to the carbon atoms on the COS backbone [33]. Compared to COS, the ^13^C NMR of COS-N exhibited peaks from 124.2 to 155.2 ppm. Specifically, the peaks at 155.2, 152.2, 135.7, 131.5 and 124.2 ppm were due to the carbons d, c, f, b and e on the pyridine ring [26]. The key peak at 165.0 ppm was ascribed to the carbon atom a on the amide bond [34]. The data above could verify the successful preparation of COS-niacin acid conjugate. After the nucleophilic substitution reaction of COS-N and BPTABs, the end products showed chemical shifts at 58.5, 19.0 and 62.2 ppm, which were the typical characteristics of carbon atoms (g, h, and i) on methylene [35]. In compound A, a peak could be found at the chemical shift of 56.5 ppm, which was the characteristic signal of the carbon atoms j on the trimethyl group. In sample B, the new chemical shift of the carbon appeared at 53.0 and 7.8 ppm, which were assigned to the carbons (j and k) on -N^+^(CH_2_CH_3_)_3_. For the spectrum of compound C, the characteristic shifts appearing at 56.5, 15.4 and 11.0 ppm were assigned to the carbon atoms (j, k, and l) of the tripropyl group. In the ^13^C NMR spectrum of compound D, the characteristic peaks at 58.6, 31.2, 19.66 and 14.0 ppm were ascribed to the carbon atoms of -N^+^(CH_2_CH_2_CH_2_CH_3_)_3_ [36]. Combining the data from the FT-IR and NMR, the results indicate the successful preparation of cationic COS derivatives bearing pyridinium and trialkyl ammonium.

### 3.4. DS Analysis

The DS of COS derivatives are shown in Table 1. It was worth noting that the DS of COS-N reached 1.80, which indicated that the conjugation between COS and nicotinic acid not only formed an amide bond at the 2-position amino group, but also an ester bond at the 6-position hydroxyl group. The results of the DS analysis were consistent with the FTIR data. For 4 end products, the COS derivatives containing trimethyl ammonium bromide had the highest DS at 0.97. Sample B presented a slightly weaker DS (0.85) and compound C possessed a DS at 0.49. The DS of COS derivatives containing tributyl ammonium bromide was relatively low at only 0.35. The reason was that the reaction efficiency was affected by the bending and folding of the longer alkyl chain.

### 3.5. Antibacterial Activity

Serious problems caused by bacterial spread are a huge health issue around the globe and arouse widespread concern, which lead to food safety issues and ultimately threaten human health [37]. To solve this problem, preparing effective antibacterial agents is of profound importance to food safety and human health. In this study, cationic COS derivatives bearing pyridinium and trialkyl ammonium were investigated for their antibacterial activity against *S. aureus*, *E. coli*, *P. citrea* and *V. harveyi*.

As can be seen in Table 2, the unmodified COS possessed no antibacterial activity with MIC and MBC values > 16 mg/mL. Compared with COS, COS-N also demonstrated an extremely weak inhibitory ability. This indicated that the introduction of the pyridine group could not significantly enhance the antibacterial activity of COS-N. After introducing BPTABs into the structure of COS by nucleophilic substitution reaction, it was apparent that the cationic COS derivatives bearing pyridinium and trialkyl ammonium had a positive effect on the antibacterial activity with MIC values between 1 and 0.125 mg/mL and MBC ranging from 8 to 0.5 mg/mL against the above pathogenic bacteria. Moreover, COS derivative A had an MIC value of 1 mg/mL and a MBC value of 8 mg/mL against *S. aureus*. Compared with sample A, the antibacterial activity of sample B had stronger antibacterial activity (MIC = 0.5 mg/mL, MBC = 1 mg/mL). Furthermore, the MIC and MBC of samples C and D were 0.5 mg/mL. The inhibition regularity of the samples against *E. coli*, *V. harveyi* and *P citrea* was similar to that of *S. aureus*. The results showed that the strong antibacterial activity was due to the introduction of the active cations and trialkyl groups. Most studies have pointed out that cationic COS derivatives exhibit biological applications involving antibacterial, antifungal and anthelmintic activities because of the electrostatic interactions. They can effectively kill bacteria through interacting with negatively charged substances in the cell membranes, such as protein and uronic acid [38]. In this study, it was obvious that the orders of antibacterial activity were ranked as follows: D > C > B > A. The antibacterial activity of the COS derivatives increased with increasing length of the alkyl chain, and sample D showed the highest efficiency. The MIC of derivatives D varied from 1 to 0.125 mg/mL and the MBC varied from 2 to 0.5 mg/mL, especially presenting a significant antibacterial effect on *V. harveyi* (MIC: 0.125 mg/mL, MBC: 1 mg/mL). Our analysis showed that the antibacterial capabilities of the COS derivatives were also related to the hydrophobic interaction of the trialkyl chain. It was suggested that the increase in the chain length strengthened the lipophilicity, which further increased the interaction between the quaternary ammonium salts and the lipid structure of the bacterial cell membranes. This changed the permeability of the bacterial cell membranes and improved the antibacterial properties [12,39]. In sum, the pyridinium salt and the quaternary ammonium salt bearing trialkyl should be excellent antibacterial function groups.

### 3.6. Antifungal Activity

*Botrytis cinerea*, *G. cingulate* and *F. oxysporum* were selected as tested fungi in this paper, and the antifungal rates of these compounds are summarized in Figure 4, Figure 5 and Figure 6. As seen in Figure 4, pure COS possessed extremely weak antifungal activity, with an inhibition rate of only 15.56% at 1.0 mg/mL against *B. cinerea*. For COS-N, it also showed weak antifungal activity at 15.26%; this finding indicates that the activity of COS-N was not significantly improved after nicotinylation. However, after introducing the pyridinium salt and alkyl groups, the inhibitory indices against *B. cinerea* of the cationic COS derivatives bearing pyridinium and trialkyl ammonium marked increasingly when compared with the pristine COS and COS-N. The inhibitory indices of samples A, B, C and D were 68.48%, 62.42%, 55.04% and 50.30% at 1.0 mg/mL, respectively, which were significantly different compared with COS and COS-N (*p* < 0.05). It was reported that pyridinium salt and quaternary ammonium salt possessed anionic-binding properties, which might interact with the negatively charged substances on the cell membrane to affect the growth of microorganisms [40]. Taking DS into account, the inhibition index of sample A with the high DS (0.97) is higher than that of sample D with the low DS (0.35) at 1.0 mg/mL. Given the effect of the difference in the DS on the compounds, we did not consider the introduction of quaternary ammonium salt and trialkyl chain-length on antifungal activity.

Figure 5 and Figure 6 showed the antifungal activity of COS and COS derivatives against *G. cingulata* and *F. oxysporum*. All end products possessed significant antifungal activity at the tested concentrations. In other words, the introduction of pyridinium salt and alkyl groups could promote the antifungal activities of COS derivatives effectively.

### 3.7. Cytotoxicity Analysis

In this article, the cytotoxicity assays of COS and COS derivatives were tested using MTT assay. As displayed in Figure 7, COS and COS-N possessed a cell viability of more than 100% at all tested concentrations. The introduction of pyridinium salt and quaternary ammonium salt containing trialkyl significantly improved the antimicrobial activity of the compounds, but their cytotoxicity was not enhanced significantly compared with COS and COS-N. The viability of L929 cells treated with the end products were 98.40%, 99.82%, 91.83% and 94.95% at 1000 μg/mL, respectively. These results proved that four end products possessed little cytotoxicity.

## 4. Conclusions

Chitooligosaccharide and its cationic derivatives were extensively researched and recognized for their enhanced antimicrobial activity. At the same time, the compounds containing alkyl groups have also attracted attention in biology and medicine because of their excellent antimicrobial ability. In the present investigation, several cationic chitooligosaccharide derivatives bearing pyridinium and trialkyl ammonium were designed, synthesized through a two-step reaction and characterized with FTIR and NMR. The results indicated that all of the obtained chitooligosaccharide derivatives possessed significantly enhanced antimicrobial activities than the pure chitooligosaccharide and chitooligosaccharide-niacin acid conjugate against four pathogenic bacteria and three harmful fungi. Specifically, all end products possessed enhanced antibacterial activity against *V. harveyi* (MIC = 0.5–0.125 mg/mL, MBC = 4–1 mg/mL), and they showed antifungal activity > 50.30% at 1.0 mg/mL against *B. cinerea*. The results above indicated that the synergy of pyridinium salt and quaternary ammonium salt containing trialkyl had a prominent increase on antimicrobial ability. In addition, the four end products possessed almost no cytotoxicity at all of the tested concentrations. Considering the above results, the obtained cationic chitooligosaccharide derivatives bearing pyridinium and trialkyl ammonium might become potential and practical candidates to be employed as a new antimicrobial reagent, which could be applied in the biological and medical fields. In addition, the antimicrobial mechanism of the compounds needs to be further researched.

## Data Availability

The data presented in this study are available on request from the corresponding author.

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
