# Peer review of "Cationic Chitooligosaccharide Derivatives Bearing Pyridinium and Trialkyl Ammonium: Preparation, Characterization and Antimicrobial Activities"

_polymers, 2022, doi:10.3390/polym15010014_

Round 1

Reviewer 1 Report

Dear authors,

Therefore, I recommend the following:

At Introduction:

The importance of obtaining these new products must be clearly emphasized.

At Conclusion:

Give a more detailed description justifying with the obtained values.

Author Response

Dear editor,

Thank you for your letter and for the reviewers’ comments concerning our manuscript entitled “Cationic Chitooligosaccharide Derivatives Bearing Pyridinium and Trialkyl Ammonium: Preparation, Characterization, and Antimicrobial Activities”. Those comments are all valuable and very helpful for revising and improving our paper. We have studied comments carefully and have made corrections which we hope meet with approval. The main corrections in the manuscript and the responds to the comments from reviewers are as following:

Responds to Reviewer

1.At Introduction: The importance of obtaining these new products must be clearly emphasized.

Answer: Thank you for your kind suggestions and according to your recommendation, the importance of obtaining these new products have been written (lines 54-57 of the revised paper).

2.At Conclusion: Give a more detailed description justifying with the obtained values.

Answer: Thank you for your kind suggestions and according to your recommendation, conclusion have been modified with detailed description (lines 416-419 and 421-422 of the revised paper).

Yours sincerely

Wenqiang Tan

Yantai Institute of Coastal Zone Research, Chinese

Academy of Sciences, Yantai, China

Email: wqtan@yic.ac.cn

Post code: 264003

Tel: +86-535-2109165

Fax: +86-535-2109000

Reviewer 2 Report

Line 137: Characterize the product. You can show it in the SI.

Line 151: what is the molar ratio between the reagents?

Line 233: the peaks at 1470 and 1500 are also observable in COS-N spectra. Why?

Line 235: The C-H groups are also present in COS and COS-N spectra. They are not new. In other words, FT-IR could not distinguish between COS-N and quaternized products.

Line 251: How did you differentiate between protons “h” and “I”?

Fig2: what is the peak at 1 ppm in COS and COS-N spectra?

Line 272: Why the COS peaks disappear in COS-N, while the DS is high. It is the same for quaternized products, too.

Line 289: Showing the spectra with integration. It can be put in SI, but necessary for the reader to see them.

Author Response

Dear editor,

Thank you for your letter and for the reviewers’ comments concerning our manuscript entitled “Cationic Chitooligosaccharide Derivatives Bearing Pyridinium and Trialkyl Ammonium: Preparation, Characterization, and Antimicrobial Activities”. Those comments are all valuable and very helpful for revising and improving our paper. We have studied comments carefully and have made corrections which we hope meet with approval. The main corrections in the manuscript and the responds to the comments from reviewers are as following:

Responds to Reviewer

  1. Line 137: Characterize the product. You can show it in the SI

Answer: Thank you for your kind suggestions and according to your recommendation, the characterization of products have been written (lines 144-154 of the revised paper).

  1. Line 151: what is the molar ratio between the reagents?

Answer: Thank you for your kind suggestions and according to your recommendation, the molar mass of COS-N has been written (lines 168 of the revised paper).

  1. Line 233: the peaks at 1470 and 1500 are also observable in COS-N spectra. Why?

Answer: Thank you for your kind suggestions, and according to your recommendation, original photographs of COS-N and A were provided in this reply. Therefore, the difference between COS-N and end products can be determined.

Fig. 1. FT-IR spectra of COS-N.

Fig. 2. FT-IR spectra of sample A.

  1. Line 235: The C-H groups are also present in COS and COS-N spectra. They are not new. In other words, FT-IR could not distinguish between COS-N and quaternized products.

Answer: Thank you for your kind suggestions. We quite agree with you, the stretching vibrations of alkyl groups could not be distinguished by FTIR. It was our negligence that led to the error. Thanks so much for your advice again.

  1. Line 251: How did you differentiate between protons “h” and “I”?

Answer: Thank you for your kind suggestions. As for sample B, the difference between protons “h” and “I” can be determined by the integral area ratio of the hydrogen signals.

About compound C and D, the difference between protons “h” and “I” can be determined by this literature.

Wang, J.; Jiang, J.; Chen, W.; Bai, Z. Synthesis and characterization of chitosan alkyl urea. Carbohydr Polym 2016, 145, 78-85.

  1. Fig2: what is the peak at 1 ppm in COS and COS-N spectra?

Answer: Thank you for your kind suggestions. COS lactate was the raw material of this study, so 1 ppm in COS and COS-N spectra was chemical shifts of the hydrogen atoms on lactate.

  1. Line 272: Why the COS peaks disappear in COS-N, while the DS is high. It is the same for quaternized products, too.

Answer: Thank you for your kind suggestions and according to your recommendation, a new figure (Fig.3. 13C NMR spectra of chitooligosaccharide and chitooligosaccharide derivatives) was redrawn (lines 283 of the revised paper). The chemical shifts of the carbon atoms on COS can be clearly seen in the Fig.3.

  1. Line 289: Showing the spectra with integration. It can be put in SI, but necessary for the reader to see them.

Answer: Thank you for your kind suggestions and according to your recommendation, a new figure (Fig.2. 1H NMR spectra of chitooligosaccharide and chitooligosaccharide derivatives) was redrawn (lines 261 of the revised paper).

Yours sincerely

Wenqiang Tan

Yantai Institute of Coastal Zone Research, Chinese

Academy of Sciences, Yantai, China

Email: wqtan@yic.ac.cn

Post code: 264003

Tel: +86-535-2109165

Fax: +86-535-2109000

Reviewer 3 Report

The manuscript shows interesting results on the antimicrobial properties of chitooligosaccharide derivatives bearing pyridinium and trialkyl ammonium. Overall the data presentation and discussions are solid and professional. I have a few minor suggestion to further improve.

1. There should be a background about why/how these particular microbes were selected for this experiment. How much they represents and how significant they are accross different applications. This can be done in introduction.

2. Though average and standard deviations are shown, the significance on the difference of data (p value) was not presented. Data of Figure 4, 5, 6 and 7 often seem overlapping each other that is why it is important to know when they were significantly different and when were not.

3. The Conclusions could be further enriched with key results by inclusion of some data.

Author Response

Dear editor,

Thank you for your letter and for the reviewers’ comments concerning our manuscript entitled “Cationic Chitooligosaccharide Derivatives Bearing Pyridinium and Trialkyl Ammonium: Preparation, Characterization, and Antimicrobial Activities”. Those comments are all valuable and very helpful for revising and improving our paper. We have studied comments carefully and have made corrections which we hope meet with approval. The main corrections in the manuscript and the responds to the comments from reviewers are as following:

Responds to Reviewer

  1. There should be a background about why/how these particular microbes were selected for this experiment. How much they represents and how significant they are accross different applications. This can be done in introduction.

Answer: Thank you for your kind suggestions and according to your recommendation, the representativeness and harmfulness of these microbes have been written (lines 39-43 and 44-47 of the revised paper).

  1. Though average and standard deviations are shown, the significance on the difference of data (p value) was not presented. Data of Figure 4, 5, 6 and 7 often seem overlapping each other that is why it is important to know when they were significantly different and when were not.

Answer: Thank you for your kind suggestions and according to your recommendation, the significantly differences of antifungal activity and cytotoxicity have been written (lines 361-372, 382-383 and 395-398 of the revised paper).

  1. The Conclusions could be further enriched with key results by inclusion of some data.

Answer: Thank you for your kind suggestions and according to your recommendation, conclusion have been modified with detailed description (lines 416-419 and 421-422 of the revised paper).

Reviewer 4 Report

The work on the “ Cationic Chitooligosaccharide Derivatives Bearing Pyridinium and Trialkyl Ammonium: Preparation, Characterization, and Antimicrobial Activities is a valuable, and suitable contribution to be published in Polymers Journal after justifying some points.

Several works were attempted to design and develop novel Chitooligosaccharide Derivatives with biological activities, the authors of the works attempt to synthesize a new series of Chitooligosaccharide derivatives that bear Pyridinium and trialkyl ammonium, and the antimicrobial and cytotoxic activities were performed accordingly, and considerable activities were collected, all of these points make this contribution valuable.

about 13 references were belongs to “Zhanyong Guo” it is better to reduce the self-citation accordingly

1-     Abstract

·       it seems the abstract should be shorter than 287 words, and the authors should focus on the main findings rather than make a long background.

·       Line 21 in vitro should be italic

·       the methods were mixed with the results, you should not mention MTT assay in the results of the abstract.

·       Remove the keyword “Quaternary ammonium salt bearing trialkyl” it seems like a sentence.

2-     Introduction

·       The introduction is well written

·       you can improve the introduction and reduce the self-citation by using recent publications regarding antimicrobial chemical structures like https://doi.org/10.1007/s13205-022-03408-8, or Arab J Sci Eng 46, 5447–5453 (2021).

3-     Materials and methods

·       you can write characterization methods instead of analytical methods, and it is better to combine FTIR and NMR in one section, no need for two separate sections.

·       Scheme 1. should be cited in the main text, and you have to present it after the cited text, it should not be in the first line of chemical synthesis.

·       Scheme 1. chemical structures’ resolution is not perfect, provide better resolution.

·       bromopropyl trialkyl ammoniums bromide can be written as an abbreviation

·       you have to citation to the chemical synthesis reaction (lines 139-159)

·       In line 166-167 the used concentration values’ presentation seems confusing with 6 decimal points, especially for the lowest concentrations.

·       to reduce the self-citation you can use this recent work according to antifungal or antimicrobial assays “Processes 2022, 10, 2050.”

4-     Results and discussion

·       This section was well written but some points should be improved or edited

·       In figure 4 the error bar of COS-N at 0.5 mg/ml concentration seems very high why ??

·       one of the major points I would like to discuss, why did you use cytotoxic concentrations in the MTT assay lower than the antimicrobial and antifungal effective concentrations ?? I think you should use very close concentrations

 Best wishes

Author Response

Dear editor,

Thank you for your letter and for the reviewers’ comments concerning our manuscript entitled “Cationic Chitooligosaccharide Derivatives Bearing Pyridinium and Trialkyl Ammonium: Preparation, Characterization, and Antimicrobial Activities”. Those comments are all valuable and very helpful for revising and improving our paper. We have studied comments carefully and have made corrections which we hope meet with approval. The main corrections in the manuscript and the responds to the comments from reviewers are as following:

Responds to Reviewer

  1. About 13 references were belongs to “Zhanyong Guo” it is better to reduce the self-citation accordingly

Answer: Thank you for your kind suggestions and according to your recommendation, new references (4, 14, 15, 16, 17, 18, 23, 24, 28, 29, 20, 41) were inserted in the article.

  1. It seems the abstract should be shorter than 287 words, and the authors should focus on the main findings rather than make a long background.

Answer: Thank you for your kind suggestions and according to your recommendation, abstract has been modified (lines 26-25 and 27-30 of the revised paper).

  1. Line 21 in vitro should be italic

Answer: Thank you for your kind suggestions and according to your recommendation, “in vitro” has been italic (line 20 of the revised paper).

  1. the methods were mixed with the results, you should not mention MTT assay in the results of the abstract.

Answer: Thank you for your kind suggestions and according to your recommendation, abstract has been modified (lines 27-29 of the revised paper).

  1. Remove the keyword “Quaternary ammonium salt bearing trialkyl” it seems like a sentence.

Answer: Thank you for your kind suggestions and according to your recommendation, Key word “Quaternary ammonium salt bearing trialkyl” has been removed (line 34 of the revised paper).

  1. You can improve the introduction and reduce the self-citation by using recent publications regarding antimicrobial chemical structures like https://doi.org/10.1007/s13205-022-03408-8, or Arab J Sci Eng 46, 5447–5453 (2021).

Answer: Thank you for your kind suggestions and according to your recommendation, new references have been cited (line 476-480 of the revised paper).

  1. You can write characterization methods instead of analytical methods, and it is better to combine FTIR and NMR in one section, no need for two separate sections.

Answer: Thank you for your kind suggestions and according to your recommendation, section 2.2 has been combined (lines 117-125 of the revised paper).

  1. Scheme 1. should be cited in the main text, and you have to present it after the cited text, it should not be in the first line of chemical synthesis.

Answer: Thank you for your kind suggestions and according to your recommendation, sections 3 have been modified (lines 237-238 of the revised paper).

  1. Scheme 1. chemical structures’ resolution is not perfect, provide better resolution.

Answer: Thank you for your kind suggestions and according to your recommendation, figure with higher resolution was re-uploaded.

  1. Bromopropyl trialkyl ammoniums bromide can be written as an abbreviation

Answer: Thank you for your kind suggestions and according to your recommendation, Bromopropyl trialkyl ammoniums bromide was written as BPTAB. (lines137, 143, 232, 253, 28, 294, and 333 of the revised paper).

  1. You have to citation to the chemical synthesis reaction (lines 139-159)

Answer: Thank you for your kind suggestions and according to your recommendation, sections 2.3 have been modified and new references have been cited (lines 137-138, 156, 168 and 471-475 of the revised paper).

  1. In line 166-167 the used concentration values’ presentation seems confusing with 6 decimal points, especially for the lowest concentrations.

Answer: Thank you for your kind suggestions. As for the concentration of each well, 100 μL of deionized water was added to each well of 96 Well Cell Culture plates, and 100μL of sample solution was added to the first well of each row, followed by diluting serially. This method allows the concentration of the first well to be twice that of the second well. Therefore, the final concentrations of the COS and its derivatives in the wells were made to be 16, 8, 4, 2, 1,0.5, 0.25, 0.125, 0.0625, 0.03125, 0.015625 and 0.0078125 mg/mL respectively.

  1. To reduce the self-citation you can use this recent work according to antifungal or antimicrobial assays “Processes 2022, 10, 2050.”

Answer: Thank you for your kind suggestions and according to your recommendation, new references were inserted in the article. References have been written (line 481-483 of the revised paper).

  1. In figure 4 the error bar of COS-N at 0.5 mg/ml concentration seems very high why ?

Answer: Thank you for your kind suggestions and according to your recommendation, the antifungal activity of COS-N against Botrytis cinerea was retested. It was our negligence that led to the error. Thanks so much for your advice again.

  1. One of the major points I would like to discuss, why did you use cytotoxic concentrations in the MTT assay lower than the antimicrobial and antifungal effective concentrations? I think you should use very close concentrations

Answer: Thank you for your kind suggestions and according to your recommendation, the cytotoxicity of chitooligosaccharide derivatives was evaluated again. Because of the slow recovery and culture of the Huvec cells, we chose L929 cells for this study. sections have been written (lines 27-30, 104, 209, 404-406 and 421-422 of the revised paper).

Round 2

Reviewer 4 Report

Dear All, 

thanks for your correction, the attached version of the manuscript, all comments were answered and edited accordingly, just i would like to ask the authors to provide better resolution of figure 2.

Best Regards